Journal of Machine Learning Research 23 (2024) 1-13        Submitted 6/24; Revised 7/24; Published 9/24

# WSI-SAM: Multi-resolution Segment Anything Model (SAM) for histopathology whole-slide images

**Hong Liu**                                                        H.LIU2@TUE.NL
*Eindhoven University of Technology*
*Eindhoven, The Netherlands*

**Haosen Yang**                                          HAOSEN.YANG.6@GMAIL.COM
*University of Surrey*
*London, UK*

**Paul J. van Diest**                                         PDIEST@UMCUTRECHT.NL
*University Utrecht*
*Utrecht, The Netherlands*

**Josien P.W. Pluim**                                              J.PLUIM@TUE.NL
*Eindhoven University of Technology*
*Eindhoven, The Netherlands*

**Mitko Veta**                                                     M.VETA@TUE.NL
*Eindhoven University of Technology*
*Eindhoven, The Netherlands*

**Editor:** My editor

## Abstract

The Segment Anything Model (SAM) marks a significant advancement in segmentation models, offering robust zero-shot abilities and dynamic prompting. However, existing medical SAMs are not suitable for the multi-scale nature of whole-slide images (WSIs), restricting their effectiveness. To resolve this drawback, we present WSI-SAM, enhancing SAM with precise object segmentation capabilities for histopathology images using multi-resolution patches, while preserving its efficient, prompt-driven design, and zero-shot abilities. To fully exploit pretrained knowledge while minimizing training overhead, we keep SAM frozen, introducing only minimal extra parameters and computational overhead. In particular, we introduce High-Resolution (HR) token, Low-Resolution (LR) token and dual mask decoder. This decoder integrates the original SAM mask decoder with a lightweight fusion module that integrates features at multiple scales. Instead of predicting a mask independently, we integrate HR and LR token at intermediate layer to jointly learn features of the same object across multiple resolutions. Experiments show that our WSI-SAM outperforms state-of-the-art SAM and its variants. In particular, our model outperforms SAM by 4.1 and 2.5 percent points on a ductal carcinoma in situ (DCIS) segmentation tasks and breast cancer metastasis segmentation task (CAMELYON16 data set). The code will be available at `https://github.com/HongLiuuuuu/WSI-SAM`.

**Keywords:** Foundation Models, Computational pathology, Whole-slide images.

## 1 Introduction

Segmentation is crucial in histopathology images, enabling pathologists to analyze tissue regions such as tumor and stroma and in turn use those results for a number of tasks such as disease diagnosis, treatment planning, and monitoring progression. Current deep learning-based models (Feng et al., 2021a; Wang et al., 2021; Xu et al., 2019; Feng et al., 2021b; Ni et al., 2019; Qaiser et al., 2019; Han et al., 2022) have shown great promise in histopathology image segmentation, which can significantly reduce time, labor, and expertise required, and enable large-scale data set analysis. However, these models are typically designed and trained for a specific segmentation task, which greatly limits the generalization to wider applications in clinical practice. Therefore, it is essential to develop universal models with zero-shot ability that can be trained once and then applied to a wide range of histopathology segmentation tasks.

Recently, the segment anything model (SAM) (Kirillov et al., 2023) was released as a segmentation foundation model for natural images, showcasing remarkable zero-shot abilities across various scenarios. Enabling its application across a wide range of tasks (Mazurowski et al., 2023; Cheng et al., 2023; Wang et al., 2023; Yue et al., 2023; Lei et al., 2023; Fazekas et al., 2023; Chen et al., 2023; Zhang et al., 2023b) through simple prompting, this break-through has catalyzed a significant paradigm shift. Some attempts have been made to apply SAM to histopathology images to boost the performance on various segmentation tasks. Med-SAM (Ma et al., 2024) fine-tunes the mask decoder of SAM on various medical data sets, while Medical SAM Adapter (Wu et al., 2023) incorporates an adapter trained on diverse medical data sets. Although these methods show strong performance on particular tasks, we hypothesize that they are suboptimal in processing histopathology WSIs that possess a pyramid structure of multiple resolutions. For instance, to capture detailed information of ductal carcinoma in situ (DCIS) lesions, patches must be extracted from images as large as $10,000 \times 10,000$ pixels at $10\times$ magnification to manage costs (van Rijthoven et al., 2021; Gu et al., 2018) and ensure compatibility with SAM, which has an input resolution of $1024 \times 1024$. A single resolution SAM model might fail because more contextual information is needed in order to "understand" the object of interest is a lesion that should also include the lesion interior, as shown in Figure 1. More visualization results can be found in the supplementary materials.

To overcome the aforementioned limitation, in this work, we propose WSI-SAM, a model that leverages multi-resolution patches (i.e., subregions from a WSI) to perform segmentation in a zero-shot manner. Similar to HookNet (van Rijthoven et al., 2021), we extract a patch from a position within the WSI and couple it with a concentric lower-resolution patch to capture essential contextual information, as illustrated in Figure 1 (right). Instead of fusing the contextual information directly at the pixel level, we execute aggregation at the token level due to token represent the segmented object information in the SAM mask generation mechanism. Given that re-training the SAM model can significantly degrades its general zero-shot performance (Ke et al., 2023), we propose the WSI-SAM architecture. This design tightly integrates with and re-uses the existing learned structure of SAM to fully retain its zero-shot capability. Specifically, we firstly introduce High-Resolution (HR) Token and Low-Resolution (LR) Token. Unlike the original output tokens, our HR and LR Token and their associated MLP layers are trained to predict masks of high-resolution and

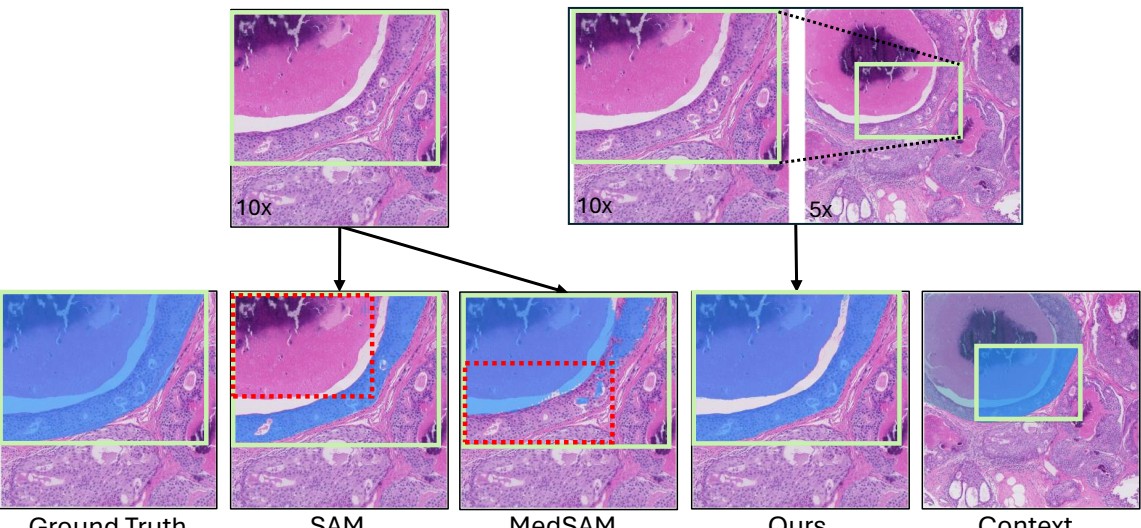

Figure 1: Comparison of DCIS segmentation in H&E-stained breast tissue by SAM, Med-SAM, and our WSI-SAM. Using the same green box as input prompt on a 10× magnification patch, SAM erroneously does not segment the interior wall of DCIS lesion. This error is compounded by the presence of calcifications and necrosis in the interior of the duct. MedSAM overlooks the ductal region beneath the lumen. Leveraging additional context (right), our WSI-SAM predict more accurate entire DCIS area, despite the intervening background and dark region.

low-resolution patches. Secondly, we propose a dual mask decoder that integrates the original SAM mask decoder with a fusion module. The fusion module enables the integration of global semantic context with local detailed features by combining SAM's mask decoder features with early and late features from its ViT encoder. Finally, instead of predicting masks independently, we integrate the HR and LR tokens at the intermediate decoder layer for contextual information aggregation across both resolutions to generate accurate mask details.

Our contributions are summarized as follows: (1) This paper introduces WSI-SAM, a segmentation framework building upon SAM, designed to seamlessly integrate contextual cues with high-resolution details, enabling the prediction of the detailed segmentation mask for histopathology images in a zero-shot manner. (2) To achieve this objective, we introduce the HR and LR Token, Dual Mask Decoder, and Token Aggregation, which together facilitate enhanced segmentation without the need for training from scratch. (3) The effectiveness of our method has been validated by two benchmarks, DCIS and CAMELYON16.

## 2 Method

### 2.1 Preliminary: SAM architecture

SAM consists of three fundamental components. The image encoder, utilizing a ViT-based backbone, extracts image features to produce image embeddings. The prompt encoder

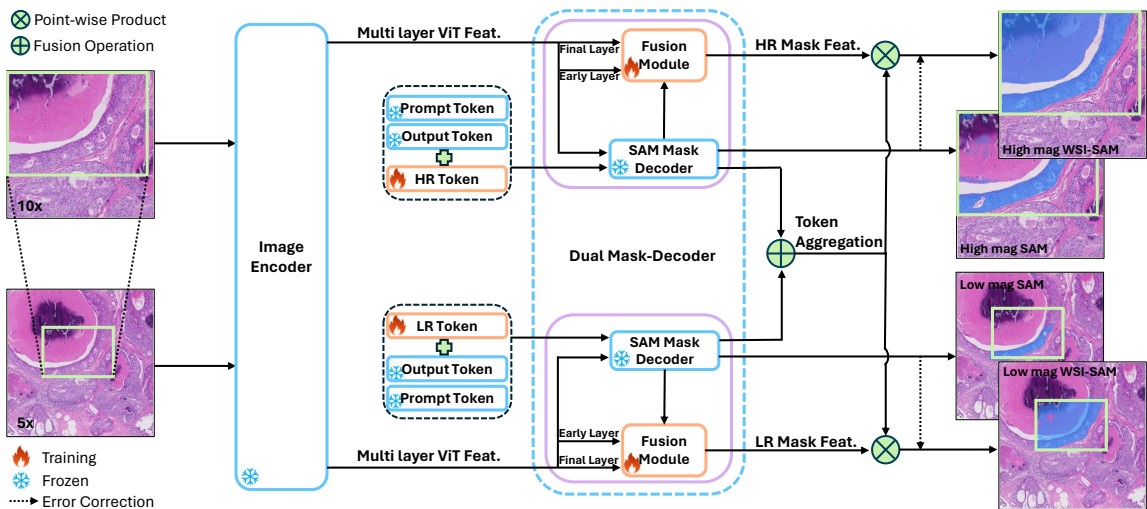

Figure 2: WSI-SAM model architecture, which introduces HR and LR Tokens, Dual Mask Decoder and Token Aggregation to SAM for for improving the mask quality in histopathology WSIs.

captures positional information from input points, boxes, or masks, aiding in the mask decoding process. Lastly, the mask decoder leverages both image embeddings and prompt tokens to generate final mask predictions. One of SAM's remarkable features is its robust zero-shot capability to novel scenarios, thanks to extensive training on a vast repository of prompt-mask pairs.

## 2.2 Ours: WSI-SAM

To maintain SAM's zero-shot transfer ability and avoid model overfitting or catastrophic forgetting (Ke et al., 2023), we opt for minimal adaptation rather than direct fine-tuning SAM or incorporating a new, extensive decoder network. To this end, we introduce three novel components in our WSI-SAM, as illustrated in Figure 2.

**High and Low Resolution Tokens** We introduce an efficient adaptation method to enhance mask quality in histopathology WSIs. As shown in Figure 2, we use the output token for mask prediction (Kirillov et al., 2023), predicting dynamic MLP weights and subsequently applying a point-wise product with the mask features. To improve mask quality in WSIs using SAM, we avoid directly utilizing SAM's coarse masks as input. Instead, we introduce HR and LR Tokens along with a new mask prediction layer, enabling refined mask prediction at multiple resolutions. As depicted in Figure 2, we preserve SAM's mask decoder unchanged while introducing two new learnable tokens, HR and LR Tokens, each with dimensions $1 \times 256$. These tokens are then merged with SAM's output tokens (sized $4 \times 256$) and prompt tokens (sized $N_{prompt} \times 256$), serving as input to SAM's mask decoder. The HR and LR Tokens interact with their corresponding resolution image features for its feature updating.

**Dual Mask Decoder**    To achieve the best interaction between token and ViT features, we propose dual mask decoder, which include the SAM mask decoder and a lightweight fusion module. HR and LR Tokens employ the point-wise MLP, which is shared with other tokens. Once processed through the mask decoder layers, the updated HR and LR Tokens acquire comprehensive insights into the global image context, crucial geometric/type information conveyed by prompt tokens, and hidden mask details inherent in other output tokens. To enhance mask quality, we augment the mask decoder features of SAM with both high-level object context and low-level boundary/edge details at Fusion modules. Specifically, we integrate new features by merging features from various stages of the SAM model, which include features from both early and late layers of ViT encoder, as well as mask features obtained from SAM's mask decoder.

**Token Aggregation**    Rather than predicting the mask independently, we introduce Token Aggregation to capture contextual information across multiple resolutions, thereby enhancing mask detail accuracy. We merge detailed features from the updated HR Token with broader contextual features from the LR Token, as these tokens represent features of the same object at different resolutions. As illustrated in Figure 2, the merge operation is formed by averaging the updated HR and LR Tokens. This aggregation method is both simple and effective, yielding segmentation results that preserve details while requiring minimal memory and computational resources. Subsequently, a spatial point-wise product is applied to the HR and LR mask features to facilitate mask generation.

### 2.3 Training Objective

While training WSI-SAM, a separate loss is computed for each resolution. We propose a loss function

$$L = \lambda L_{high} + (1 - \lambda)L_{low},$$

where $L_{high}$ and $L_{low}$ are the combination of dice loss and cross-entropy loss for different resolutions, respectively, and $\lambda$ controls the importance of each resolution.

## 3 Experiments and Results

**Training data**    To train WSI-SAM in a data-efficient manner, we train on the **CATCH** (Wilm et al., 2022) data set. This data set contain 350 WSIs of seven distinct subtypes of canine cutaneous tumors, augmented with 12,424 polygon annotations across 13 histological classes. Following the official division, the data set is split into a training set with 245 WSIs and a validation set encompassing 35 WSIs.

**Benchmark settings**    We assess our model's performance on two Whole Slide Image (WSI) data sets in a zero-shot manner. The first is a data set of Ductal Carcinoma in Situ (**DCIS** (Wetstein et al., 2021)), a non-invasive breast cancer collection comprising 116 WSIs. From this, 50 WSIs were randomly selected for our test set. The second is the **CAMELYON16**[1] data set, where we incorporated its official test set into our own. This set includes a total of 47 WSIs.

---

1. `https://camelyon17.grand-challenge.org/Data/`

## 3.1 Implementation Details

During training, we maintain the model parameters of the pre-trained SAM model unchanged, focusing solely on making the proposed WSI-SAM learnable. Given SAM's capability for handling flexible segmentation prompts, we train WSI-SAM using a variety of prompt types, such as bounding boxes, randomly sampled points, and coarse masks. These degraded masks are created by incorporating random Gaussian noise into the boundary areas of the GT masks. Given hardware constraints, we randomly select a limited number of $1024 \times 1024$ patches at $10\times$ magnification and their corresponding concentric patches at $5\times$ magnification from each WSI for training. We adpot TinyViT (Zhang et al., 2023a) as its backbone. Our training employs a mini-batch size of 1 and a learning rate 0.001. We conducted all experiments using PyTorch on a setup equipped with an NVIDIA TITAN Xp GPU.

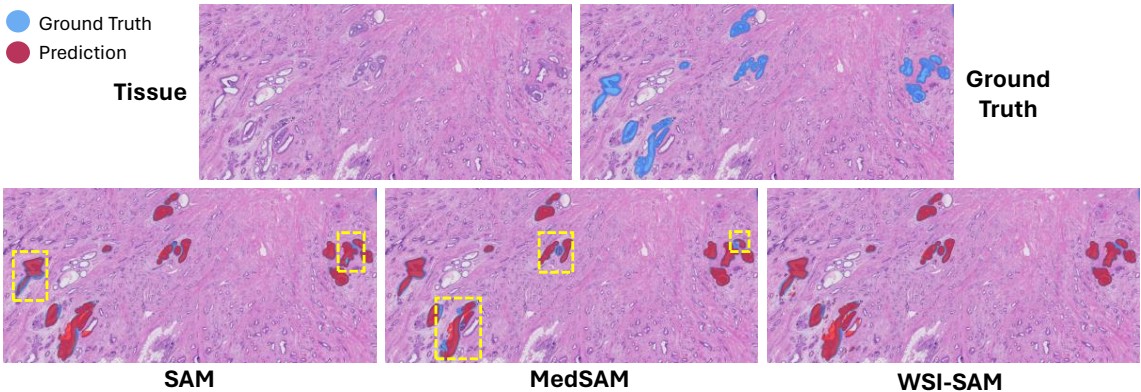

Figure 3: Example of predicted tissue segmentation on DCIS. Yellow boxes indicate the incorrect predictions.

## 3.2 Zero-shot Comparison with SAMs

**Setup** In our evaluation process, we conduct zero-shot mask prediction using prompts. Our investigation includes two distinct types of prompts: box and point. To accurately assess improvements in mask quality, we employ the Dice Similarity Coefficient (DSC) as our reporting metric, which quantifies the overlap between the predicted and ground truth masks, providing a reliable measure of segmentation performance.

**Zero shot segmentation with box prompt** We compare with SAM, fine-tuned SAM and MedSAM on both DCIS and CAMELYON16 data sets. We simulate realistic human-annotated bounding boxes by introducing noise to the GT object boxes. As shown in Table 1, we observe a marked decrease in performance when fine-tuning SAM on histopathology WSIs, likely due to model overfitting or catastrophic forgetting during the fine-tuning process (Ke et al., 2023). In contrast, WSI-SAM outperforms SAM by 4.1 and 2.5 percent points on the DCIS and CAMELYON16 data sets, respectively. Furthermore, when compared to MedSAM, which was trained on a variety of medical data sets, WSI-SAM demonstrates a

| Method | DCIS | CAMELYON16 | | | |
|---|---|---|---|---|---|
| | DCIS | IDC | ILC | DUC | avg |
| SAM (Kirillov et al., 2023) | 73.35 | 86.06 | 81.36 | 72.86 | 80.09 |
| SAM* | 61.22 | 74.17 | 72.88 | 40.00 | 62.35 |
| MedSAM (Ma et al., 2024) | 64.71 | 77.91 | 75.24 | 71.64 | 74.93 |
| WSI-SAM (Ours) | **77.50** | **89.76** | **84.30** | **73.55** | **82.54** |

Table 1: Comparison of zero-shot segmentation results on the DCIS and CAMELYON16 test sets using bounding boxes as input prompts. * indicate fine-tune SAM's mask decoder on CATCH.

| Method | SAM | MedSAM | WSI-SAM |
|---|---|---|---|
| DSC | 56.00 | 56.43 | **57.37** |

Table 2: Comparison of zero-shot segmentation results on the test set of DCIS. We use nnU-Net (Isensee et al., 2021) trained on DCIS as box prompt generator.

significant performance enhancement, as evidenced by the increase from **64.71%** to **77.50%**. Moreover, to simulate real-world scenarios, we further evaluated our method using prompts generated from nnU-Net (Isensee et al., 2021). WSI-SAM surpasses both SAM and Med-SAM (e.g., **56.0, 56.73** vs **59.9**), showcasing the resilience and robustness of our approach. To provide additional validation, we conduct a segmentation qualitative analysis on DCIS, as illustrated in Figure 3.

**Zero shot segmentation with point prompt**   To investigate the segmentation capabilities of WSI-SAM using interactive point prompts, we conducted a comparison with SAM, evaluating their performances across a range of input point quantities on both the DCIS and CAMELYON16 data sets. Noting that MedSAM is limited to box prompts. For the CAMELYON16 data set, we present an average performance across the three tumor classes, with comprehensive metrics for each class detailed in the supplementary materials. WSI-SAM consistently outperforms SAM on both data sets, regardless of the number of point prompts utilized, as shown in the Figure 4.

### 3.3  Ablation Study

We conducted an ablation study for WSI-SAM on DCIS data set with bounding boxes as input prompts.

**Effect of the HR and LR Tokens.**   WSI-SAM leverages HR and LR Tokens to fuse more contextual information, enhancing mask prediction accuracy. We examined various aggregation target, specifically fuse the HR and LR feature by average-pooling, fuse with expand HR feature region on the HR feature, and HR and LR token aggregation. Results presented in Table 3b indicate that utilizing HR and LR Tokens, outperforms these alternatives, achieving performance improvements of 5.3 percent points on DCIS data set.

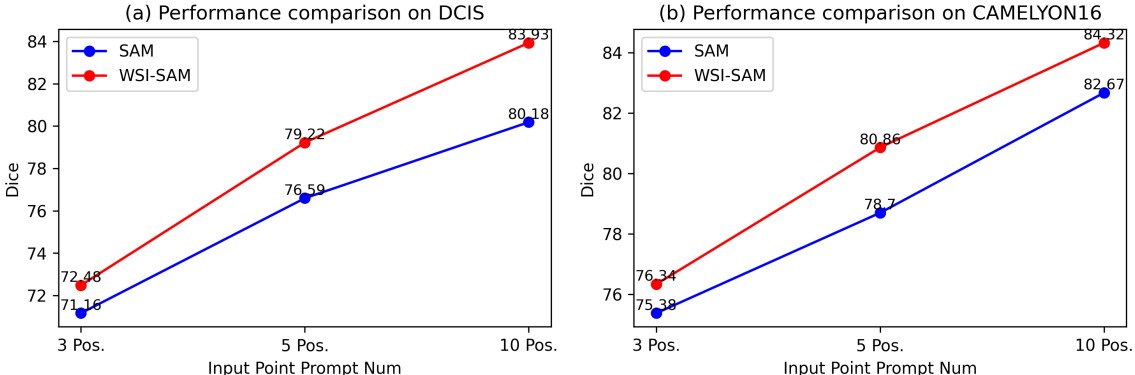

Figure 4: Comparison of zero-shot interactive segmentation results using a varying number of input points on the DCIS and CAMELYON16 data set.

| Aggregation | Dice |
|---|---|
| Concat.-FC | 77.24 |
| Max. | 75.91 |
| Avg. | **77.50** |

(a)

| Aggregation Target | Dice |
|---|---|
| HR feat. and LR feat. | 72.20 |
| HR feat. and expand HR feat. | 59.04 |
| HR Token and LR Token | **77.50** |

(b)

| $\lambda$ | Dice |
|---|---|
| 0.0 | 74.16 |
| 0.5 | **77.50** |
| 1.0 | 75.20 |

(c)

Table 3: Ablation experiments on DCIS data set. (a) Ablation study on aggregation ways of HR and LR features or Tokens using box prompts; (b) Ablation study on aggregation target for integrating contextual information using box prompts; (c) Ablation study on different values of $\lambda$ in loss function.

**Ablation on Tokens Aggregation** We evaluate three ways of fusing HR and LR Tokens: (1) Concatenating followed by one learnable FC layer (Concat-FC); (2) Max pooling (Max); (3) Average pooling (Avg). Table 3a shows that the average pooling turns out to be the best way for aggregation.

**Ablation on $\lambda$ in loss function.** The influence of losses from HR and LR Tokens is modulated by $\lambda$. Experiments were conducted with $\lambda = 0.0$ to ignore the HR Token, $\lambda = 0.5$ to balance both tokens equally, and $\lambda = 1.0$ to ignore the LR Token. According to Table 3c, setting $\lambda = 0.5$ yields the best performance enhancement.

## 4 Conclusion

We introduce WSI-SAM, the zero-shot segmentation model tailored for WSIs, incorporating innovative HR and LR Tokens to refine the mask prediction of SAM's output token and enhancing the original SAM with minimal additional computational cost. Our zero-shot transfer evaluations on the DCIS and CAMELYON16 data sets showcase WSI-SAM's superior performance, marking a substantial advancement in zero-shot segmentation for WSIs.

**Appendix A.**

| Method | IDC | ILC | DUC |
|--------|-----|-----|-----|
| SAM | 85.26 | 80.76 | 60.13 |
| WSI-SAM | **85.63** | **81.66** | **61.73** |

(a)

| Method | IDC | ILC | DUC |
|--------|-----|-----|-----|
| SAM | 87.53 | 82.09 | 66.48 |
| WSI-SAM | **89.53** | **84.90** | **68.16** |

(b)

| Method | IDC | ILC | DUC |
|--------|-----|-----|-----|
| SAM | 89.43 | 83.97 | 74.61 |
| WSI-SAM | **91.53** | **86.23** | **75.22** |

(c)

Table 4: Comparison of zero-shot segmentation results on CAMELYON16 test sets using different numbers of points as input prompts. (a) 3 positive points; (b) 5 positive points; (c) 10 positive points.

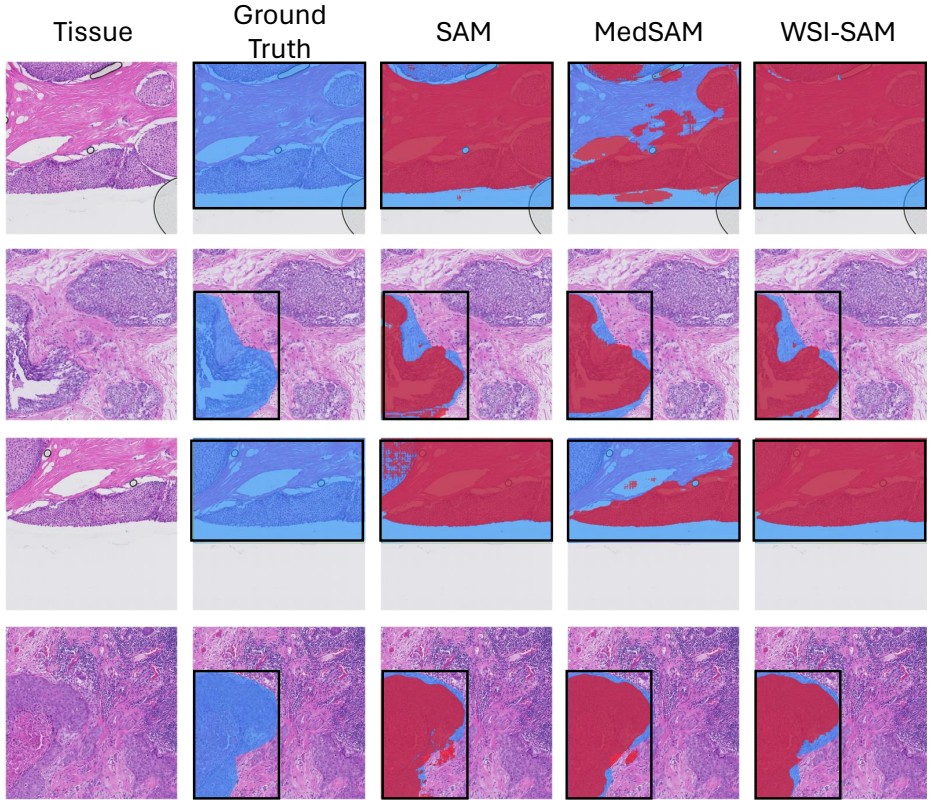

Figure 5: Comparison of segmentation mask predictions in DCIS data set.

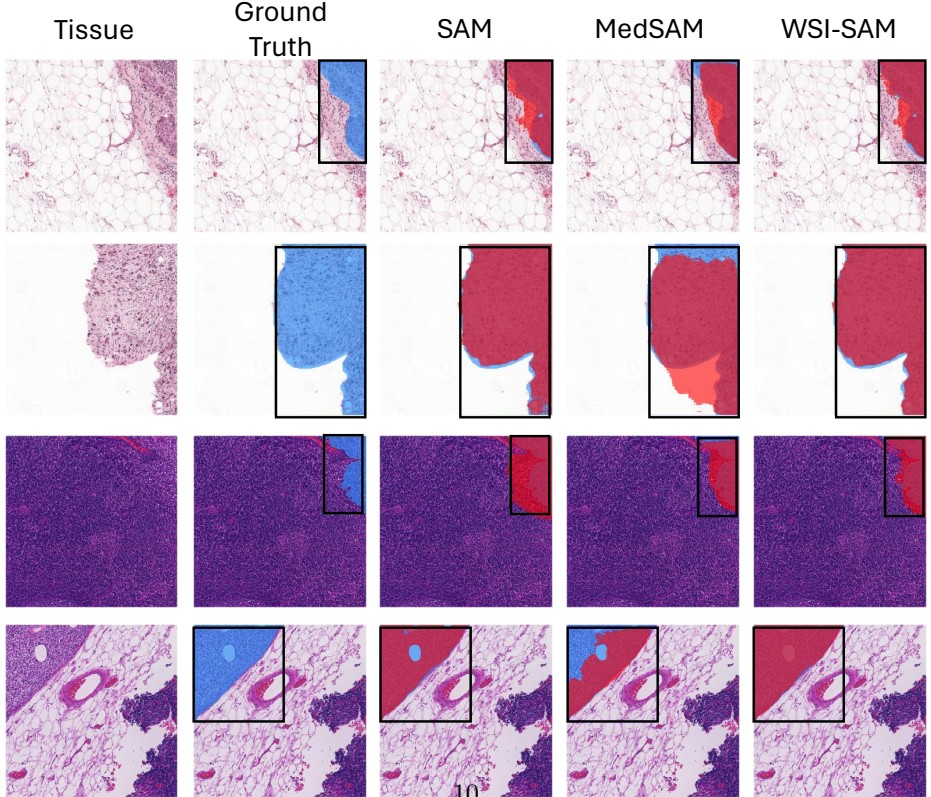

Figure 6: Comparison of segmentation mask predictions in CAMELYON16 data set.

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
