# OpenReview forum: "WSI-SAM: Multi-resolution Segment Anything Model (SAM) for histopathology whole-slide images"
_MICCAI.org/2024/Workshop/COMPAYL — COMPAYL 2024_

### Official Review · Reviewer_83dU · 2024-07-02
**A great multi-resolution upgrade for SAM to segment WSIs**

**Custom Rating:** 5
**Confidence:** 5

**Review:**

The authors proposed WSI-SAM by enhancing SAM with precise object segmentation capabilities for histopathology images using
multi-resolution patches, while preserving its efficient, prompt-driven design, and zero-shot abilities.

Pros:

1. Neat improvement of well known SAM with multi-resolution approach;
2. Tested on a well known CAMELYON16 dataset with great results for WSI-SAM;

Cons:

Quite minor - it's hard to see the differences in the approaches in Figure 3 unless zooming in significantly. Maybe focus on 1-2 regions and provide larger images?

Also, just curious, if you had a chance to explore HR / LR as a hyperparameter or any maybe you could add a sentence to guide the reader regarding how to choose HR/LR and whether adding more resolutions could help even further?

Finally, what are computational costs of adding the two resolutions to SAM?

---

### Official Review · Reviewer_5kVW · 2024-07-08
**Well written paper with good experimental setup and validation**

**Custom Rating:** 5
**Confidence:** 4

**Review:**

Summary of paper:

The paper is tackling the issues of the segment anything model in histopathology by creating a multi-resolution SAM that is merging features from high and low resolution for better mask predictions. They demonstrate superior performance compared to both the original SAM model and domain specific MedSAM and include ablation studies to experimentally verify their approach.

Pros:
- The manuscript is clear and well written
- The proposed method is easy to understand and implemented in a straightforward and elegant way.
- There are a lot of experiments and included ablation studies to support the proposed method

Major concerns:
-  None

Minor concerns:
- For clarity, please add information the evaluation metric(s) used in Table 1
- Adding a boundary-based metric such as Hausdorff distance would make the manuscript more convincing, particularly because in the examples provided, the distance to boundary is more relevant than the area.
- in "Ablation on λ in loss function" you are only disabling the loss during training of WSI-SAM but not disabling the feature itself because the weights would still be initialized randomly. Therefore, the text sounds like a false description? Please clarify.

Other comments:

- For citing, could you use another \cite variant or add brackets around the citations? The currently used format makes it hard to read as the citations seem to be part of the sentence.
- "Moreover, to simulate real-world scenarios, we further evaluated our method using prompts generated from nnU-Net"
Could you elaborate on how this is supposed to simulate real world scenarios?
- It would be nice if the manuscript could be re-read to fix some grammar and incomplete sentences (e.g. "datasets. Noting that MedSAM is limited to box prompts. For ")
- Its curious that SAM performs better than MedSAM in the evaluated tasks. Did you investigate why that is? If you have some insights, those would be great to include.

Code, model weights, and/or data availability:
- github repository available but nothing pushed at the time of review

Conclusion

The proposed multiresolution WSI-SAM is a great addition to the SAM model family and the manuscript is well written with good experimental setup.

---

### Official Review · Reviewer_Abkf · 2024-07-11
**Review of paper 5: a segmentation framework based on SAM**

**Custom Rating:** 5
**Confidence:** 3

**Review:**

The authors introduced a segmentation framework based on SAM, which integrates contextual cues with high-resolution details to accurately predict detailed segmentation masks for histopathology images. The results demonstrate that this proposed framework surpasses the classic SAM and its variations.

Pros:
- The method was tested across various experiments and high-quality datasets.

Cons:
- The authors should include a section for their evaluation metrics, explaining the metrics used and their justification for choosing them.
- Currently, it is not clear how the method was evaluated with the metrics. Did the author do any type of cross-validation or are the results reported only on the test set?

Minor comment:
- Some text sections with citations need to be rewritten for clarity, as they are currently difficult to follow.

---

### Decision · Program_Chairs · 2024-07-16

Accept